# Vet-ICD-O-Canine-1, a System for Coding Canine Neoplasms Based on the Human ICD-O-3.2

**DOI:** 10.3390/cancers14061529

**Published:** 2022-03-16

**Authors:** Katia Pinello, Valeria Baldassarre, Katja Steiger, Orlando Paciello, Isabel Pires, Renée Laufer-Amorim, Anna Oevermann, João Niza-Ribeiro, Luca Aresu, Brian Rous, Ariana Znaor, Ian A. Cree, Franco Guscetti, Chiara Palmieri, Maria Lucia Zaidan Dagli

**Affiliations:** 1Departamento de Estudo de Populações, Vet-OncoNet, ICBAS, Instituto de Ciências Biomédicas Abel Salazar, Universidade do Porto, 4050-313 Porto, Portugal; jjribeiro@icbas.up.pt; 2EPIUnit—Instituto de Saúde Pública, Universidade do Porto, 4050-600 Porto, Portugal; 3Laboratório Para a Investigação Integrativa e Translacional em Saúde Populacional (ITR), 4050-600 Porto, Portugal; 4Department of Veterinary Medicine and Animal Production, University of Naples Federico II, 80138 Naples, Italy; valeria.baldassarre@unina.it (V.B.); paciello@unina.it (O.P.); 5Institute of Pathology, School of Medicine, Technical University of Munich, 81675 Munich, Germany; katja.steiger@tum.de; 6Associate Laboratory for Animal and Veterinary, Science-AL4AnimalS Animal and Veterinary Research Centre (CECAV), University of Trás-os-Montes and Alto Douro (UTAD), 5001-801 Vila Real, Portugal; ipires@utad.pt; 7School of Veterinary Medicine and Animal Science, São Paulo State University (UNESP), Botucatu 18618-681, SP, Brazil; renee.laufer-amorim@unesp.br; 8Division of Neurological Sciences, DCR-VPH, Vetsuisse Faculty, University of Bern, 3012 Bern, Switzerland; anna.oevermann@vetsuisse.unibe.ch; 9Department of Veterinary Sciences, University of Turin, 10095 Grugliasco, Italy; luca.aresu@unito.it; 10National Disease Registration Service, NHS Digital, London SE1 8UG, UK; brian.rous@nhs.net; 11International Agency for Research on Cancer, IARC, 69372 Lyon, France; znaora@iarc.fr (A.Z.); creei@iarc.fr (I.A.C.); 12Institute of Veterinary Pathology, Vetsuisse Faculty, University of Zurich, 8057 Zurich, Switzerland; franco.guscetti@vetpath.uzh.ch; 13School of Veterinary Science, Gatton Campus, The University of Queensland, Gatton, QLD 4343, Australia; 14School of Veterinary Medicine and Animal Science, University of São Paulo, São Paulo 05508-270, SP, Brazil; mlzdagli@usp.br

**Keywords:** cancer registry, canine, coding, comparative oncology, ICD-O-3.2

## Abstract

**Simple Summary:**

The development of a widely accepted and comparable animal cancer registration system lacks standardized animal cancer coding. The GIVCS group have developed a comparative coding system for canine neoplasms—Vet-ICD-O-canine-1—compatible with the human ICD-O-3.2 and consistent with the currently recognized classification schemes for canine tumors. This system comprises 335 topography codes and 534 morphology codes allowing the collection of consistent epidemiologic canine cancer data and offering a robust framework for comparative oncology studies.

**Abstract:**

Cancer registries are fundamental tools for collecting epidemiological cancer data and developing cancer prevention and control strategies. While cancer registration is common in the human medical field, many attempts to develop animal cancer registries have been launched over time, but most have been discontinued. A pivotal aspect of cancer registration is the availability of cancer coding systems, as provided by the International Classification of Diseases for Oncology (ICD-O). Within the Global Initiative for Veterinary Cancer Surveillance (GIVCS), established to foster and coordinate animal cancer registration worldwide, a group of veterinary pathologists and epidemiologists developed a comparative coding system for canine neoplasms. Vet-ICD-O-canine-1 is compatible with the human ICD-O-3.2 and is consistent with the currently recognized classification schemes for canine tumors. It comprises 335 topography codes and 534 morphology codes. The same code as in ICD-O-3.2 was used for the majority of canine tumors showing a high level of similarity to their human counterparts (*n* = 408). De novo codes (*n* = 152) were created for specific canine tumor entities (*n* = 126) and topographic sites (*n* = 26). The Vet-ICD-O-canine-1 coding system represents a user-friendly, easily accessible, and comprehensive resource for developing a canine cancer registration system that will enable studies within the One Health space.

## 1. Introduction

According to the Global Cancer Observatory (GCO) from the International Agency for Research on Cancer (IARC) of the World Health Organization (WHO), the cancer burden in humans is progressively increasing worldwide. The same trend is observed among companion animals, with over 4.2 million dogs diagnosed with cancer annually [1]. Dogs are considered unique models of human cancer because, among others, they develop cancers with similar morphological and biological features and share the same environment as humans [1]. This makes them promising translational models for human cancer therapy and potential sentinels of environmental exposure to carcinogenic agents [1]. Therefore, canine cancer data may provide important information in a comparative oncology and One Health approach.

Cancer data are the cornerstone for cancer prevention and control, and cancer registries represent fundamental tools for collecting validated and comprehensive data [2]. The establishment and maintenance of human cancer registries have been complex and time-consuming [2]. Currently, human cancer registries are mandatory and one in three countries report high-quality cancer incidence data [2]. Initiatives to set up veterinary cancer registries have been developed since the early 1960s [3,4,5,6], however most have been discontinued for reasons including the non-mandatory nature of animal cancer case reporting, limited funding, and challenges in enumerating the background population. Despite the fact that only a few veterinary cancer registries are currently active, we are witnessing a revamped interest in the topic worldwide [7,8,9].

A pivotal aspect in the development of a widely accepted and comparable cancer registration system is the availability of standards for cancer coding [4,8,10]. The specific issue of coding was addressed in 1976, when the WHO published the first edition of the International Classification of Diseases for Oncology (ICD-O) [11]. The second edition, published in 1990, was developed by a WHO/IARC working party that was also involved in the refinement of the current third edition-second revision (ICD-O-3.2) that provides a standardized coding system for the topography (site), morphology (histologic diagnosis or nomenclature of the tumor), and behavior of cancers [11].

Although few studies have used the ICD-O-3.2 for coding animal tumors [5,12], the lack of codes for specific veterinary entities has hindered the consistent and valid collection of cancer data within and across countries and species. In response to the urgent need for an international consensus and standardized guidelines for animal cancer registration [3,4], the Global Initiative for Veterinary Cancer Surveillance (GIVCS) was established in 2018 [8] to set up guidelines for coding animal cancers and to facilitate communication and collaboration between different veterinary cancer registries. Thereafter, a subgroup consisting of veterinary pathologists and epidemiologists from six countries engaged in the development of a tumor coding system for use in animal cancer registration. Major requirements for such a system were (a) the highest adherence to currently recognized classification schemes for animal tumors, and (b) the highest level of similarity and compliance with the human topography and morphology codes.

This work provides a standardized coding system and practical guidelines for coding canine tumors, allowing the collection of consistent epidemiologic canine cancer data and offering a robust framework for comparative oncology studies.

## 2. Materials and Methods

### 2.1. Workflow for Establishing the Veterinary Coding System

The GIVCS veterinary coding subgroup consists of nine veterinary pathologists and two veterinary epidemiologists from six countries (Portugal, Brazil, Australia, Italy, Germany, and Switzerland) who reviewed the current canine tumor classification systems and discussed the criteria and the procedures to fit the cancer nomenclature into an ICD-O-3.2-compatible coding system.

The 12 organ systems defined in the current WHO subdivision (Table 1) were assigned to individual veterinary pathologists who generated draft proposals of lists of tumors that were discussed by the whole group. Subsequently, each system underwent a detailed review process within small working groups and then each system was revised to ensure consistency. Finally, IARC scientists (IAC, BR, AZ) revised the work to ensure the highest level of standardization and compliance with the human coding system. The veterinary coding system presented here was specifically developed for canine tumors and named the Veterinary International Classification of Diseases for Oncology Canine Tumors First Edition (Vet-ICD-O-canine-1).

### 2.2. Definition of Items for the Vet-ICD-O-Canine-1 Lists

The Vet-ICD-O-canine-1 code lists were created using the ICD-O-3.2 lists (with American spelling) as templates. Terms were either included or excluded based on their adherence to the current veterinary classification terminology. Both the topographic and morphologic lists follow the numerical order of the codes, as in the ICD-O-3.2.

#### 2.2.1. Topographic List

The list included the level (1, organ; 2, topography item), the Vet-ICD-O-canine-1 code, a description of the topographic site and subsite, the included topographies, excluded topographies, and notes. Codes denoting an identical topographic location named differently in humans vs. animals (e.g., arms in humans and forelimbs in dogs) were retained and the topographic term was replaced.

#### 2.2.2. Morphologic List

The list included the ICD-O-3.2 code, the Vet-ICD-O-canine-1 code, the morphologic group (e.g., epithelial neoplasms, adenomas, adenocarcinomas), the veterinary term (name/s of the tumor according to the current veterinary classification schemes), the level (preferred, synonym, related), and the topography (for entities related to a specific topography).

Out-of-date terminology was retained for some tumor entities to allow the correct coding of retrospective and historical data and assigned the [obs] (obsolete) descriptor.

#### 2.2.3. List of Tumors Grouped by Organ Systems

Each organ system was structured in a separate table including the items’ ICD-O-3.2 code, Vet-ICD-O-canine-1 code, veterinary term, level, topography, and notes/sources (issues arising in the applicability of the code, rationale for the introduction of new codes, and sources not included in the main reference list).

Within the organ systems list, the entities were grouped and ordered according to the WHO International Histological Classification of Tumors of Domestic Species and the first four new surgical pathology fascicles from the David Thompson Foundation [13,15,17,18,19,24,30,32,34,36,38,40,41,42], complemented with updated information from additional textbooks [14,26,27,43] and relevant publications. Case reports were considered when deemed to contain sufficient information to define a new neoplastic entity. As an exception, the coding of central nervous system tumors was based on the morphological criteria of the 2016 human WHO classification [23,44], considering the obsolete WHO classification of the tumors of the nervous system of domestic animals, which dates back to 1999 [41], and the current alignment of leading veterinary neuropathologists with the human WHO classification [22]. Some degree of incongruency with the current ICD-O-3.2 coding scheme for tumors of the nervous system became evident, and the authors defined the different entities in a way that would allow the highest degree of comparison with the human coding and classification system. Therefore, all existing ICD-O-3.2 codes were retained if they were denoting entities recognized in dogs.

For lymphatic malignancies, the authors mostly relied on a publication by Valli et al. (2011) [28] reporting an updated summary table of the Revised European-American Lymphoma REAL/WHO classification, as well as on Meuten’s report [14]. The terms of the updated Kiel classification, currently in use by some veterinary pathologists, were added if deemed appropriate [45]. Additional terms mentioned in the last WHO classification of canine lymphoid tumors dating back to 2002 [43] and adapted from the human 2008 4th edition WHO classification [46], which derives from the REAL [47], were retained [43].

For cutaneous mast cell tumors (MCTs), two different grading systems are commonly used (Patnaik vs. Kiupel), either individually or in combination [48,49,50]. However, the sixth digit of the ICD-O-3.2 morphology code containing the grading information was not considered suitable for capturing all clinically relevant combinations of grades. Therefore, as an exception, new morphological codes were created to indicate different combinations of grades in cutaneous MCTs.

Regarding the level category, the preferred morphologic terms were derived from the nomenclatures used in the canine WHO classification or the most used terms for a particular neoplastic entity. Synonyms were chosen from frequently used alternative terms designating the same or a very similar entity; they may help the registrars to find the correct morphology code when a term other than the preferred one is used. Related terms were selected among terms representing variants or subtypes of the entity indicated by the preferred term or equivalent terms adopted from an alternative classification system.

### 2.3. Structure of ICD-O-3.2 Codes

To align the veterinary cancer coding system with human coding, and thus satisfy the dual purpose of creating a solid framework for animal cancer surveillance studies and allow comparative oncology research, the guidelines and rules set by the ICD-O-3.2 were adopted [11].

In the ICD-O-3.2, the topography codes describe the site of origin of the neoplasm. Topography terms have four-digit codes running from C00.0 to C80.9. Decimal point(s) separate subdivisions (subsites) of the specified topography [11]. For example, the *urinary bladder neck* is coded C67.5, with C67 indicating the urinary bladder (site) and .5 the subsite.

The morphology codes describe tumor characteristics, such as cell type and biological behavior [11]. Morphology terms have four-digit codes ranging from 8000 to 9993. The first four digits indicate the specific histological type. A fifth digit, which follows the slash or stroke (/), is a behavior code which indicates whether a tumor is benign (/0), uncertain whether benign or malignant (/1), a carcinoma in situ (/2), malignant (/3), metastatic (/6), or uncertain whether a primary or metastatic site (/9) [11].

An additional one-digit code is provided for histologic grading (differentiation) (codes 1–4 and 9, for malignant tumors) and immunophenotype (codes 5–9, for lymphomas and leukemias) [11]. For example, a *well-differentiated transitional cell carcinoma* would be coded as M-8120/31 with 8120 denoting the histologic type (transitional cell neoplasm), 3 the biological behavior (malignant), and 1 the histologic grade (well differentiated). A complete ICD-O code thus requires 10 digits or characters [11] to identify the topographic site (four characters), morphologic type (four digits), behavior (one digit), and grade/immunophenotype (one digit) [11].

### 2.4. Structure of Vet-ICD-O-Canine-1 Codes and Criteria for Their Assignment

In the Vet-ICD-O-canine-1, codes were either adopted from the ICD-O-3.2 or created de novo by the authors with a structure making them unique and recognizable (Table 2). Specifically, the new veterinary topography codes contain a second digit after the dot (at position 5), while the new morphology codes were generated by adding a dot and a digit after position 4. The new veterinary codes were set close to the most similar/related ICD-O-3.2 codes to facilitate aggregation.

The rules to assign morphologic codes in the Vet-ICD-O-canine-1 are listed in Table 3. The following solutions were applied to three different scenarios:(1)Codes were fully adopted from the ICD-O-3.2 and used in the same way in the Vet-ICD-O-canine-1.For most tumors, the human and canine classifications largely overlap, thus allowing the allocation of the exact same code with the same structure as in the ICD-O-3.2 (Table 2). For example, a urothelial carcinoma of the urinary bladder neck would be coded in dogs, as in humans, as C67.5 (urinary bladder neck), M-8120/3 (urothelial carcinoma). With tumors showing a different behavior in dogs vs. humans, the behavior code (digit after the slash) was adapted accordingly.(2)Codes were adopted from the ICD-O-3.2 and used in the Vet-ICD-O-canine-1 to characterize tumors and histological subtypes not included in the ICD-O-3.2.If a histologic subtype, which can be defined using a pre-existing ICD-O-3.2 generic code (used to indicate that specific histologic subtype in different organs or systems), was not listed in the human classification but present in the canine classification, then those generic codes were used. One typical example is the subtype of adenocarcinoma, whose codes can be applied to different organ systems, with the topography code being critical in separating them. For example, adenocarcinoma of the nasal cavities is classified in humans as an intestinal type (coded as 8144/3) or non-intestinal type, with the latter showing a predominant papillary and/or tubular growth pattern. In the ICD-O-3.2, these morphological subtypes (papillary and tubular) are not individually coded; instead, all subtypes of the non-intestinal type are assigned the generic (i.e., not specific to the nasal cavities) adenocarcinoma NOS code (8140/3). In dogs, three different histologic subtypes can be recognized; therefore, three generic codes (8260/3, papillary adenocarcinoma; 8263/3, tubulopapillary carcinoma; 8550/3, acinar adenocarcinoma) have been introduced in addition to code 8140/3.(3)Codes were created de novo.New codes were needed for the following: (a) canine-specific entities or anatomic structures not included in the human classification system (e.g., 9375.1/0, canine transmissible venereal tumor; C60.31, os penis); (b) histologic subtypes, which could not be defined using a generic term from the ICD-O-3.2, but should be distinguished in veterinary medicine pending validation through prognostic studies (e.g., 8650.1/1 interstitial cell tumor, diffuse-compact; 8650.2/1 interstitial cell tumor, vascular-cystic; 8650.3/1 interstitial cell tumor, pseudoadenomatous); (c) entities with a similar or identical term in both classification systems but showing different histologic features, grading, and/or clinical manifestations in the two species (e.g., 9740.2/1-9740.9/1 for different grades of mast cell tumors); and (d) entities indicated as related in the ICD-O-3.2 but occurring with a much higher frequency in dogs.


## 3. Results

The Vet-ICD-O-canine-1 comprises a topography list with 335 topography codes and includes 26 new veterinary topographies (Appendix A). The morphology list comprises 534 codes (preferred terms), with 126 being de novo created veterinary codes (Appendix A). There are also 248 synonyms and 64 related terms, for a total of 846 terms.

The following sections refer to the lists of tumors grouped by organ systems (Appendix A) and indicate the main features of the new topographic and morphologic codes and related issues.

### 3.1. Tumors of Bone, Cartilage, and Other Hard Tissues

Most entities share terminology with the human classification system. New codes were introduced for *multilobular tumor of the bone* (9210.1/3), *maxillary andmandibular fibrosarcoma* (8812.2/3), *central chondrosarcoma* (9220.1/3), *central fibrosarcoma* (8812.1/3), *amyloid-producing ameloblastoma* (9310.1/0), *ameloblastic carcinoma* (9270.1/3), and *odontogenic sarcoma* (9270.2/3). Considering the differences in the classification of osteosarcoma in humans and dogs, new codes were created for *osteoblastic* (9180.1/3), *giant cell* (9180.2/3), *poorly differentiated* (9180.3/3), and *peripheral osteosarcoma* (9192.0/3), which are not included in the ICD-O-3.2.

### 3.2. Tumors of Soft Tissue

*Soft tissue tumor* (STT) is a commonly used term in diagnostic practice in dogs including tumors with different histogenesis and biological behaviors, such as fibrosarcoma, perivascular wall tumors, nerve sheath tumors, and undifferentiated sarcomas. A more accurate characterization usually requires ancillary tests and/or special stains, which are not always performed. In addition, there are insufficient data to predict the outcome and biological behavior of many canine STTs. Therefore, a new veterinary code was created for *soft tissue tumor*, *NOS* (8800.0/1), to differentiate this group of neoplasms from *sarcoma, NOS* (8800/3), and to avoid confusion regarding their potential aggressiveness and prognoses (Appendix A).

### 3.3. Epithelial and Melanocytic Tumors of the Skin

The *nail bed* is a recognized site of origin for specific tumors in dogs. Therefore, new topographic codes were created for the *nail bed of the forelimbs* (C44.61), *nail bed of the hindlimbs* (C44.71), and the *nail bed, NOS* (C44.92). In addition, a new code for the *skin of the limbs, NOS* (C44.91), was created for tumors without specification into the fore- or hindlimbs (Appendix A).

New codes were created for the histologic subtypes of *basal cell carcinoma*, *solid* (8090.1/3) and *clear cell* (8090.2/3); *tricholemmoma*, *isthmic* (8102.1/0) and *inferior* (8102.2/0); *trichoblastoma, ribbon* (8105.2/0), *medusoid* (8105.3/0), *granular cell* (8105.4/0), *trabecular* (8105.5/0), *spindle cell* (8105.6/0), *solid/cystic* (8105.7/0); *meibomian adenoma* (8410.1/0) and *adenocarcinoma* (8410.1/3); *hepatoid adenoma* (8410.2/0), *epithelioma* (8410.2/1), and *adenocarcinoma* (8410.2/3). Finally, a new code for *melanocytoma* (8720.0/0) was introduced to distinguish this benign entity from *melanoma*, *NOS* (8720/3) (Appendix A).

### 3.4. Tumors of the Genital Tract

To enable the coding of neoplastic entities described only in dogs, additional topographic codes were included for the *canine vestibule* (C52.10), *ovarian bursa* (C56.10), *rete testis* (C62.81), and *os penis* (C60.31) (Appendix A).

*Canine transmissible venereal tumor*, another tumor entity unique to dogs, was assigned a separate code in the group of miscellaneous tumors (9375.1/0). New codes were assigned to the histologic subtypes of *fibroleiomyoma of the vagina* (8890.1/0) and of *seminoma* (9061.1/1 *intratubular,* 9640.2/1 *diffuse*), *Sertoli cell tumor* (8640.1/1 *intratubular*, 8640.2/1 *diffuse*), and *interstitial cell tumor* (8650.1/1 *diffuse-compact*, 8650.2/1, *vascular-cystic*, with the synonym *angiomatoid*, and 8650.3/1 *pseudoadenomatous*) (Appendix A).

### 3.5. Tumors of the Nervous System

A total of 23 new codes were generated (i) to reflect the granularity of the human 2016 WHO classification system [23] and the growing numbers of CNS tumor subtypes described in dogs [14], (ii) to account for the revised classification of canine gliomas [22], and iii) to establish a comprehensive list of codes for all relevant subtypes and grades. For example, in the case of astrocytomas, new separate codes for *diffuse* (9400.0/3), *compact low-grade* (9400.1/3), *focally infiltrative low-grade* (9400.2/3), *compact anaplastic* (9401.0/3), and *focally infiltrative anaplastic astrocytoma* (9401.1/3) were added to the pre-existing ICD-O-3.2 code (9400/3) for *astrocytoma, NOS* (Appendix A). Additionally, a new code was created for the *ectopic nephroblastoma of the canine thoracolumbar spinal cord*, a specific canine entity related to nephroblastoma, which is not described in humans.

### 3.6. Tumors of the Respiratory System

The major changes applied to the topographic coding system compared to the ICD-O-3.2 included: (1) the use of *paranasal sinuses* as a preferred term for C31 instead of the synonym, accessory sinuses; (2) the different terminologies for the lung lobes that in animals are indicated as the *cranial* (C34.1), *middle* (C34.2), and *caudal lobe* (C34.3); (3) the inclusion of a new code for the *accessory lobe* (C34.40); and (4) the different terminologies for the mediastinum, subdivided into *cranial* (C38.1) and *caudal mediastinum* (C38.2) (Appendix A).

Tumors of the respiratory system in dogs show a high level of similarity with the corresponding entities in humans, and only minor changes were made with respect to the ICD-O-3.2 coding. A new code was assigned under the category of miscellaneous tumors to *combined carcinoma of the lung* (9376.1/3). This tumor, described by Meuten [14], could not be matched with any of the variants of pulmonary carcinoma in humans (Appendix A).

### 3.7. Tumors of the Hematopoietic System

Because of the variety of terms used to designate individual lymphoma entities, the ICD-O-3.2 frequently provides one or several synonyms. In the present work, synonyms were added restrictively when there was reasonable evidence that they would allow the correct assignment of subtypes defined in former classification schemes. As Hodgkin’s lymphomas are not mentioned in the current canine classification scheme, the ICD-O-3.2 code for *malignant lymphoma, NOS* (9590/3), was not retained in the Vet-ICD-O-canine-1. Instead, the code for *malignant lymphoma (non-Hodgkin), NOS* (9591/3) was retained for non-classified lymphomas, i.e., canine lymphomas diagnosed without immunophenotyping (unless the morphology is unequivocal). Due to the common use of cytology and immunocytology for diagnosing lymphoma in dogs, a new code was introduced for *B-cell lymphoma, NOS* (9591.1/3). Follicular lymphoma (FL) rarely occurs in dogs, and the current human grading system is not directly translatable to dogs [28], resulting in its very rare use. In addition, the current understanding is that FL progresses from grade one to three over time and the grading is equivalent to a staging evaluation [27]. Therefore, the authors propose to group all FL subtypes defined by grade in the ICD-O-3.2 under the term *follicular lymphoma*, *NOS* (9690/3). The entity *T-zone lymphoma* was assigned a new veterinary code (9702.1/3) because of its higher relative frequency in dogs compared to humans (resulting in its listing as a synonym of *mature T-cell lymphoma*, *NOS* (9702/3) in the ICD-O-3.2) (Appendix A). As a recommendation, the five most common canine lymphoma subtypes, which should be coded whenever possible, comprise (1) 9680/3, *diffuse large B-cell (non-cleaved, cleaved) lymphoma*, *NOS*; (2) 9689/3, *splenic marginal zone B-cell lymphoma*; (3) 9699/3, *marginal zone B-cell lymphoma*, *NOS* (which includes the nodal subtype and the mucosa-associated lymphoid tissue subtype); (4) 9702/3, *peripheral T-cell lymphoma*, *NOS* (the preferred name in ICD-O-3.2, mature T-cell lymphoma, NOS); and (5) *9702.1/3*, *T-zone lymphoma*.

#### Mast Cell Tumors (MCTs)

There is a poor overlap between mast cell tumors in dogs and humans. The human classification is based on additional molecular and clinical criteria and the actual ICD-O-3.2 code structure adds a further level of complexity. MCTs are routinely graded and, as an exception, new codes (9740.1/1 through 9740.9/1) were created to indicate different clinically relevant combinations of grades (Appendix A).

Cutaneous and subcutaneous MCTs in dogs have different behaviors [14]; therefore, a separate code number was assigned to the latter (9740.0/1) (Appendix A).

### 3.8. Ocular and Otic Tumors

The anatomical location of the *third eyelid* or *nictitating membrane* was assigned a new code (C69.01). Similarly, new codes were assigned for the *meibomian gland adenoma* (8410.1/0), *epithelioma* (8410.1/1), and *adenocarcinoma* (8410.1/3), neoplasms occurring in dogs that are not coded in humans (Appendix A).

### 3.9. Tumors of the Alimentary System

With respect to topography, there are some major differences between dogs (and other animals) and humans in the alimentary system. *Glandular* and *non-glandular compartments of the stomach* (C16.41, resp.C16.42) were added as topography codes in the Vet-ICD-O-canine-1 (Appendix A). The classification and coding of the neoplastic entities of the alimentary system in dogs resemble the human classifications.

### 3.10. Tumors of the Urinary Tract

Among the organ systems, the urinary tract exhibited the highest degree of similarity between human and dog coding. Only one new code was introduced in the Vet-ICD-O-canine-1, namely for the *nonpapillary urothelial carcinoma*, *infiltrating* (8120.1/3) variant (Appendix A).

### 3.11. Mammary Gland Tumors

New veterinary topographic codes for each canine mammary gland *cranial thoracic* (C50.10), *caudal thoracic* (C50/20), *cranial abdominal* (C50.30), *caudal abdominal* (C50.40), and *inguinal* (C50.50)) were created, since the distinction in portions and quadrants of the human breast is not equivalent to the number and site of canine mammary glands (Appendix A).

In the histological classification of canine mammary tumors, some entities are unique to dogs and are not listed in the human breast cancer classification system. In the following cases, the authors introduced new specific veterinary codes for *carcinoma arising in a complex adenoma/benign mixed tumor* (8941.1/3) and *complex carcinoma* (8983.1/3) (Appendix A).

### 3.12. Tumors of the Endocrine System

The classification and coding of the tumors of the endocrine system in dogs parallels the ICD-O-3.2, except for the distinction between *compact cellular carcinoma* (8330.1/3) and *mixed compact follicular-cellular carcinoma* (8330.2/3) of the thyroid gland, and *chromophobe ACTH-secreting adenoma* (8270.1/0) and *chromophobe inactive adenoma* (8270.2/0) of the pituitary gland. In addition, new codes were created for the canine entities *malignant mixed thyroid tumors* (8330.3/3), *thyroglossal duct remnant tumor* (8260.1/3), and *pancreatic polypeptidoma* (8150.1/1) (Appendix A).

## 4. Discussion

This study provides a reference system for coding canine tumors for cancer registration purposes and for facilitating comparative oncology studies, epidemiologic investigations, and tumor mapping. The uniformity and standardization offered by the Vet-ICD-O-canine-1 aids in overcoming the inconsistencies and lack of coordination inherent to previous initiatives of veterinary cancer registration. Adhering to the present coding system will enable a reliable comparison of registry data from different countries and allow for large-scale epidemiologic studies on canine cancer.

The Vet-ICD-O-canine-1 is consistent with the ICD-O-3.2 and most morphological terms and codes overlap fully, as expected given the well-described histologic similarities between human and canine neoplasms. However, the new coding system considers the peculiarities of some canine tumors, resulting in 126 new codes for entities that cannot be reliably coded with the ICD-O-3.2.

The use of new codes reflects the need to capture individual histologic subtypes and the variety of presentations in dogs in the most comprehensive and detailed manner. This will provide greater flexibility for later adjustments and code conversions and allow future large-scale correlation studies, including molecular and clinical data, prognostic prospective and retrospective analyses, and epidemiologic investigations. The growing interest in the identification of genomic and transcriptomic signatures of canine tumors will undoubtedly lead to the creation of new neoplastic entities, mostly based on their molecular characteristics, which will become important prognostic, predictive, and therapeutic biomarkers, as already observed in human oncology. The fine granular coding adopted in the Vet-ICD-O-canine-1 will constitute a solid basis for further expanding the coding system based on such characteristics. Despite the introduction of new codes, interspecies comparisons will be possible to a large extent, since the codes created de novo were numerically set as close as possible to the related human entities, thus enabling aggregation of codes for comparative studies. The authors understand that tumor classification in domestic animals is constantly evolving, and new entities will be described after the publication of the Vet-ICD-O-canine-1. Minor modifications will be included in the tables available online on the GIVCS website (https://www.givcs.org, accessed date 9 March 2022) with the indication of the date of editing. Major changes may result in future releases following new releases of the ICD-O-3.2 or updates of the canine WHO tumor classification, if deemed appropriate. If new entities are described in dogs but are not yet captured in the Vet-ICD-O-canine-1, the authors recommend contacting the GIVCS or the corresponding author to discuss the creation of new codes.

## 5. Recommendations for the Use of Vet-ICD-O-Canine-1

The authors propose the following recommendations for the most appropriate application of the Vet-ICD-O-canine-1.

### 5.1. Criteria for Assigning the Biological Behavior Code (Fifth Digit in ICD-O-3.2)

Although six behavior codes are defined in the ICD-O-3.211, the behavioral codes assigned in the reference tumor lists of both the ICD-O-3.2 and Vet-ICD-O-canine-1 only comprise codes /0, /1, /2, and /3 (indicating benign, uncertain whether benign or malignant, carcinoma in situ, and malignant, respectively). They are consistent with the most common behavior of each specific cancer type according to the latest histological classification [14,17,19,22,24,26,30,32,34,36,38,40,41,43,51,52,53,54]. However, in a practical coding situation, the behavior code should be adjusted if a different behavior is observed in a particular case. In addition, to include information regarding metastatic lesions in the canine population, the authors encourage the use of codes /6 and /9 by animal cancer registries. This is a significant deviation from the current coding practices in human cancer registries. Because of their focus on primary tumors in order to describe the burden of cancer and inform cancer control at the population level, human cancer registries focus only on the primary site and refrain from using codes /6 (malignant, metastatic site) and /9 (malignant, uncertain whether primary or metastatic site) [11,55]. If the site of the primary cancer is unknown and cannot be determined based on the morphology code, this information is conveyed by the topographic code C80.9 (unknown primary site) in the ICD-O-3.2.

In dogs for example, sebaceous epithelioma (8410/1) is commonly associated with a good prognosis after complete surgical excision. However, in the presence of a recurrence or a metastatic lesion of a sebaceous epithelioma, the behavior code should be changed to /3 or /6, respectively. As another example, carcinoma spreading to the lung from an unknown site of origin should be coded as C34.6 (lung) or 8010/9 (carcinoma/unknown if primary or metastatic). A metastatic carcinoma in an intra-abdominal lymph node would be coded as C77.2, 8010/6 (carcinoma NOS, metastatic). If a malignant neoplasm is suspected to be metastatic and not primary, in the absence of clinical information, or if clinical data support a metastatic condition, the behavior code /9 should be used. If the behavior is unclear or not stated, it should be coded according to the Vet-ICD-O-canine-1. Table 4 describes the morphology code matrix concept that should be applied to veterinary coding systems.

### 5.2. Grading Code (Sixth Digit)

The ICD-O-3.2 includes a separate one-digit code for the histologic grading or differentiation and immunophenotyping of lymphomas and lymphatic leukemias. The authors discourage the use of this code at present because the usefulness and translational value of many proposed grading systems are still debated [56,57] and information about the degree of differentiation or grading may not be consistently reported. This is in line with the current practice of human coding.

It is beyond the scope of this work to suggest the application of certain grading systems (except for mast cell tumors and meningiomas). However, to ensure consistency the authors recommend assigning the 6th digit code when available and desired as follows: (1) 2-tier grading systems: assign code 1 for low grade, and code 3 for high grade; (2) 3-tier grading systems: assign code 1 for grade I, code 2 for grade II and code 3 for grade III. For example, a grade II tubular carcinoma of the mammary gland in dogs according to the 3-tier grading system proposed by Goldschmidt et al. [37] should be optionally coded as 8211/32. Mentioning the grading system is also desirable. If the degree of differentiation is mentioned instead of grading, the following sixth digit code should be optionally applied according to the definition in the ICD-O-3.2: code 1 for well-differentiated, code 2 for moderately differentiated, code 3 for poorly differentiated, code 4 for undifferentiated/anaplastic. When a diagnosis indicates two different grades or degrees of differentiation, the highest number should be registered [11]. Regarding the sixth digit code for immunophenotyping lymphomas and leukemias, the four-digit histology code of the tumors of the hematopoietic system included in the Vet-ICD-O-canine-1 already contains the cell lineage; therefore, an additional digit is not required.

### 5.3. Criteria for Using Terms with the Descriptor NOS (Not Otherwise Specified)

The authors have applied the descriptor “NOS” in the tumor lists according to the rules set by the ICD-O-3-2, i.e., for entities where one or more modifiers have been indicated. The codes of terms with the descriptor NOS should be selected:1.When the pathology report contains a generic diagnosis without additional modifying words. For example, adenomas or carcinomas without further indication of histologic types (that appear elsewhere in the Vet-ICD-O-canine-1 lists) must be coded as adenoma, NOS (8140/0), and adenocarcinoma, NOS (8140/3);2.When in the pathology report a modifier not listed in the Vet-ICD-O-canine-1 is used. An atypical adenocarcinoma is coded 8140/3 (adenocarcinoma, NOS) since the adjective atypical is not included in the list of modifiers for adenocarcinoma;3.In borderline cases where the distinction between two tumors or subtypes is not obvious. For example, a urothelial carcinoma arising from the canine prostatic urethra cannot be differentiated from a prostatic carcinoma of glandular origin. Therefore, in borderline cases where the distinction between glandular and urothelial carcinoma of the prostate cannot be made, the code 8010/3 (carcinoma, NOS) should be applied.

## 6. Conclusions

The Vet-ICD-O-canine-1 represents a user-friendly, easily accessible, and comprehensive resource for veterinary pathologists, epidemiologists, and cancer researchers working in the canine tumor registration field. Its creation represents an important milestone towards the standardization of canine cancer registration, allowing comparability between existing and future cancer registries at the intra- and interspecies levels.

## Figures and Tables

**Table 1 cancers-14-01529-t001:** Sequential list of tumor groups ordered by organ systems and main references used for compiling the respective tables.

	Organ Systems	Main References
1.	Tumors of bone, cartilage and other hard tissues	[13,14]
2.	Tumors of soft tissue	[14,15]
3.	Epithelial and melanocytic tumors of the skin	[14,16,17,18]
4.	Tumors of the genital tract	[19,20]
5.	Tumors of the nervous system	[14,21,22,23]
6.	Tumors of the respiratory system	[24,25]
7.	Tumors of the hematopoietic system	[14,26,27,28,29]
8.	Ocular and otic tumors	[30,31]
9.	Tumors of the alimentary system	[32,33]
10.	Tumors of the urinary tract	[34,35]
11.	Mammary gland tumors	[36,37]
12.	Tumors of the endocrine system	[38,39]

**Table 2 cancers-14-01529-t002:** Structure of the codes used in the Vet-ICD-O-canine-1.

Code Type	ICD-O-3.2 Codes ^1^	New Vet-ICD-O-1 Codes ^2^
Topography	C_ _._ *site.subsite*	C_ _._ _ *site.**subsite vet***
Morphology	(M-) __ __ __ __/__ __ *Histology Behavior Grade*	(M-) __ __ __ __. __/__ __ *Histology Vet Behavior Grade*

^1^ For entities largely overlapping between humans and dogs, the ICD-O-3.2 codes were adopted. ^2^ For the remaining entities, new veterinary codes containing the added features highlighted in gray were created.

**Table 3 cancers-14-01529-t003:** Rules applied for the definition of the Vet-ICD-O-canine-1 morphologic codes.

	Item	Action
1.	Tumors described in both humans and dogs with similar histologic features.	ICD-O-3.2 codes are retained in the Vet-ICD-O-canine-1.
2.	Similar tumors in both species with different preferred terms.	The ICD-O-3.2 codes and the veterinary preferred term are retained.
3.	Tumors not described in dogs.	ICD-O-3.2 codes are excluded from the Vet-ICD-O-canine-1.
4.	Tumors showing a different behavior in dogs compared to humans.	The code for behavior (the digit after the slash) is changed.
5.	Histologic subtypes in specific organs in dogs which can be defined using a generic term but are not listed in the human classification for that specific organ.	Generic codes indicating histologic subtypes are used.
6.	Tumors described only in dogs.	Codes are created de novo according to the structure as outlined in Table 2, right column.
7.	Histologic subtypes unique to dogs or not specifically coded in humans.	Codes are created de novo according to the structure outlined in Table 2, right column.

**Table 4 cancers-14-01529-t004:** Morphology and behavior code matrix.

	Example A	Example B
Basic Cell Type	8140	8150
**5th digit, Behavior code**		
/0 Benign	8140/0	8150/0
	Adenoma, NOS	Islet cell adenoma
/1 Uncertain whether benign or malignant	8140/1	8150/1
	Bronchial adenoma	Pancreatic endocrine tumor, NOS
/2 In situ; non-invasive	8140/2	
	Adenocarcinoma in situ	
/3 Malignant	8140/3	8150/3
	Adenocarcinoma, NOS	Pancreatic endocrine tumor, malignant
/6 Malignant, metastatic	8140/6	8150/6
	Adenocarcinoma, metastatic	Pancreatic endocrine tumor, metastatic
/9 Malignant, uncertain whether primary or metastatic	8140/9	
	Adenocarcinoma (primary or metastatic)	

NOS: Not Otherwise Specified.

## Data Availability

The data that supports the findings of this study are available in the Appendix A of this article.

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
