# Peer review of "Vet-ICD-O-Canine-1, a System for Coding Canine Neoplasms Based on the Human ICD-O-3.2"

_cancers, 2022, doi:10.3390/cancers14061529_

Round 1

Reviewer 1 Report

Dear Authors, we thank you for submitting your work to us, the subject matter is of great interest and certainly well conducted. I congratulate you on your valuable and useful initiative.

Author Response

Thank you very much for taking time in reviewing our manuscript. We are delighted to see that the manuscript has been favorably accepted and we are looking forward to providing this important coding system to the broader scientific community.

Reviewer 2 Report

The authors developed and described a novel animal cancer registration system, namely Vet-ICD-O-canine-1, based on the human ICD-O-3.2 cancer coding system. This is a great system that would allow to collect and standardize canine cancer data, that is not only important for veterinary epidemiological studies, but could also contribute to comparatively strengthen the relationship between human and canine cancers, within the One Health era. The study is very well done, and it is evident the authors put a lot of efforts in developing this canine cancer registration system, which is thoroughly described and gives a powerful tool to clinicians, oncologists, pathologists, epidemiologists, and other veterinary specialists.

The only major concern refers to how user-friendly this system will be right now. I understand that every single tumor entity will be specifically coded, based on topography, morphology, and other features, such as grading. As a result, each canine tumor entity is assigned to a specific code. But how do the authors see this applicable in practice? In other words, who assigns this code to the specific cancer entity? I believe that one strategy to make this system easy to use and even more user-friendly would be to create a web-based interface where, for example, an oncologist, or a pathologist can access, select the cancer entity from pre-made lists, and eventually get the automatically generated specific code. In this way, the code is generated by the software, rather than created by the user, who could possibly make mistakes and report the wrong code.

Another point I would like to bring to the authors’ attention, just as a comment, is why previously proposed animal cancer registries have been discontinued? And in what is the authors’ proposed system improved from the previously proposed systems?

More specific/minor comments:

The authors suggest using the sixth digit code “/1” for grade I, and code “/3” for grade II for those tumors with 2-tier grading systems and I understand why. However, I find that a bit confusing considering that grade II and grade III tumors that have 3-tier grading systems would be assigned to code “/2” and “/3”, respectively. Again, if there was a web-based interface that would automatically generate the code, this would not be an issue. However, if the code was generated by the user instead, I personally would find this confusing as one would be more inclined to give the code “/2” to grade II tumors, regardless of whether they have a 2-tier or 3-tier grading system.

Lines 57-60: please rephrase.

Lines 140-142: please rephrase. Do the authors mean “List of items that includes level, Vet-ICD-O-canine-1 code, ….”?

Lines 147-150: please rephrase. Do the authors mean “List of items that includes ICD-O-3.2 code, Vet-ICD-O-canine-1 code, ….”?

Please double-check the supplementary table for inconsistencies. For example, on the “Morphology list” sheet, in rows 220, 221, 222, under the “Topography” column, I believe the authors meant “C73.-“, instead of “C73-“. Also, sometimes an “underscore” digit is used after the dot, sometimes a “dash” digit is used. Please choose either of the two and modify accordingly, to be consistent. Please check all the other sheets.

Line 175 and 180: the “REAL” acronym should be explained the first time it appears, so in line 175, and not in line 180.

Line 183: Do the authors mean “six-digit numbers”, instead of “sixth digit numbers”? If so, please modify accordingly also elsewhere.

Line 511: what does “6” stand for? I believe the authors meant “assign code 1 for Grade I, code 2 for Grade II, and code 3 for grade III.”

Author Response

Comments and Suggestions for Authors
The authors developed and described a novel animal cancer registration system, namely VetICD-O-canine-1, based on the human ICD-O-3.2 cancer coding system. This is a great system that would allow to collect and standardize canine cancer data, that is not only important for veterinary epidemiological studies, but could also contribute to comparatively strengthen the relationship between human and canine cancers, within the One Health era. The study is very well done, and it is evident the authors put a lot of efforts in developing this canine cancer registration system, which is thoroughly described and gives a powerful tool to clinicians, oncologists, pathologists, epidemiologists, and other veterinary specialists.

The only major concern refers to how user-friendly this system will be right now. I understand that every single tumor entity will be specifically coded, based on topography, morphology, and other features, such as grading. As a result, each canine tumor entity is assigned to a specific code. But how do the authors see this applicable in practice? In other words, who assigns this code to the specific cancer entity? I believe that one strategy to make this system easy to use
and even more user-friendly would be to create a web-based interface where, for example, an oncologist, or a pathologist can access, select the cancer entity from pre-made lists, and eventually get the automatically generated specific code. In this way, the code is generated by the software, rather than created by the user, who could possibly make mistakes and report the wrong code.

Thank you very much for taking time in reviewing our manuscript and for your invaluable comments and suggestions that have definitely improved the quality of our study. Regarding the creation of a web-based interface, this is something that we are currently exploring. The main scope of our study was to create a coding system that pathologists, researcher and epidemiologists can apply as soon as the publication is made available to the broader community.
One of the aims of this international collaboration is to provide guidelines for the registration of tumors in domestic animals in a user-friendly manner but we understand this is a long process of which the creation of a specific veterinary coding system not available before, was the first necessary step. The interface as you mentioned above is out of the scope of our study, but we are currently exploring the opportunity to build an open access, user-friendly system similar to the CanReg5 in consultation with business analysts and software engineers.
We are also conscious that this process must take in account the differences between human and veterinary medicine and also the differences existing between the different realities at a geographical level. A system that requires advanced economical or technological resource could discourage someone to use it.
Providing data collection methods, inclusion and exclusion criteria and definition of the reference population that will generate a cancer registration system is the goal of GIVCS but as first step our priority was to create a veterinary coding system and we believe that excel file, at least in this first moment, could be more feasible for the majority of the veterinary realities.
The Excel file is actually a pre-made list of morphological diagnoses and corresponding codes that, depending on the resource of each institution, could be integrated in their database in a way that searching for a morphological diagnosis a code is automatically assigned.

Another point I would like to bring to the authors’ attention, just as a comment, is why previously proposed animal cancer registries have been discontinued? And in what is the authors’ proposed system improved from the previously proposed systems?

Thank you again for your observation that helped us to better explain the rationale of our work. The discontinuity of previous animal cancer registries is mainly related to lack of funds and financial support, but also the lack of a specific veterinary coding system led the institutions involved in animal cancer registration to use human coding system without a standardized approach.
There was no previously proposed system, or in other words, the only system was the human ICD-O. However, the latter is based on the human classification systems that is not completely transferrable to the veterinary tumor classification system, and that resulted in the application of the human coding mainly according to the subjective choice of pathologist or oncologist or epidemiologist involved in assigning the codes.
The existence of some morphological, molecular and clinical differences between human and animal cancer, makes critical to adopt a different classification system, and a different classification system implies a different coding system. This is the first tumor coding system based on canine classification of tumors.
In this Vet-ICD-O many codes were adapted from the human ICD-O but in an unprecedented multidisciplinary effort involving people from a variety of background ranging from veterinary pathology and epidemiology to human pathology and epidemiology.

More specific/minor comments:
The authors suggest using the sixth digit code “/1” for grade I, and code “/3” for grade II for those tumors with 2-tier grading systems and I understand why. However, I find that a bit confusing considering that grade II and grade III tumors that have 3-tier grading systems would be assigned to code “/2” and “/3”, respectively. Again, if there was a web-based interface that would automatically generate the code, this would not be an issue. However, if the code was
generated by the user instead, I personally would find this confusing as one would be more inclined to give the code “/2” to grade II tumors, regardless of whether they have a 2-tier or 3-tier grading system.

Thank you very much for this observation that allowed us to find an error in the paper that now we can correct.
We totally agree with your point and we realized that our sentence was very confusing.
The scope of a 2-tier grading system is to divide tumor in two group always referred as low and high grade and there is no grading system (at least in veterinary oncology) that use a 2-tier grading system dividing tumors in I and II grade.
We changed it in the paper, and we believe that re-phrasing the sentence as: “1) 2-tier grading systems: assign code 1 for low Grade, and code 3 for high Grade” should help to avoid any confusion.
Lines 57-60: please rephrase.
Rephrased accordingly.
Lines 140-142: please rephrase. Do the authors mean “List of items that includes level, VetICD-O-canine-1 code, ….”?
Changed accordingly.
Lines 147-150: please rephrase. Do the authors mean “List of items that includes ICD-O-3.2 code, Vet-ICD-O-canine-1 code, ….”?
Changed accordingly.
Please double-check the supplementary table for inconsistencies. For example, on the “Morphology list” sheet, in rows 220, 221, 222, under the “Topography” column, I believe the authors meant “C73.-“, instead of “C73-“. Also, sometimes an “underscore” digit is used after the dot, sometimes a “dash” digit is used. Please choose either of the two and modify accordingly, to be consistent. Please check all the other sheets.
Thank you for your careful revision. We changed it accordingly.
Line 175 and 180: the “REAL” acronym should be explained the first time it appears, so in line
175, and not in line 180.
Thank you for your careful revision. We changed it accordingly.
Line 183: Do the authors mean “six-digit numbers”, instead of “sixth digit numbers”? If so,
please modify accordingly also elsewhere.
Thank you. We change to “sixth digit” and deleted “numbers”.
Line 511: what does “6” stand for? I believe the authors meant “assign code 1 for Grade I, code
Line 511: what does “6” stand for? I believe the authors meant “assign code 1 for Grade I, code
2 for Grade II, and code 3 for grade III.”
Thank you. “6” was a typo. It was deleted. Changed accordingly

Reviewer 3 Report

Dear Authors, congratulations for the quality of the work that will contribute to the internationalization of the veterinary coding system.

The uniformity and standardization presented in this manuscript could overcome the lack of coordination inherent to previous initiatives of vet cancer registration.

Author Response

Thank you so much for reviewing and positively judging our work. We believe that it is an important milestone in achieving a more global and standardized application of cancer registration and coding in veterinary medicine.